# Patient satisfaction with hospital care and nurses in England: an observational study

Linda H Aiken,[1] Douglas M Sloane,[1] Jane Ball,[2] Luk Bruyneel,[3] Anne Marie Rafferty,[4] Peter Griffiths[2]

[1]School of Nursing, University of Pennsylvania, Center for Health Outcomes and Policy Research, Philadelphia, Pennsylvania, USA
[2]Faculty of Health Sciences, University of Southampton, Southampton, UK
[3]University of Leuven, Leuven Institute for Healthcare Policy, Leuven, Belgium
[4]Florence Nightingale School of Nursing and Midwifery, King's College London, London, UK

**Correspondence to**
Professor Linda H Aiken;
laiken@nursing.upenn.edu

## ABSTRACT

**Objectives** To inform healthcare workforce policy decisions by showing how patient perceptions of hospital care are associated with confidence in nurses and doctors, nurse staffing levels and hospital work environments.

**Design** Cross-sectional surveys of 66 348 hospital patients and 2963 inpatient nurses.

**Setting** Patients surveyed were discharged in 2010 from 161 National Health Service (NHS) trusts in England. Inpatient nurses were surveyed in 2010 in a sample of 46 hospitals in 31 of the same 161 trusts.

**Participants** The 2010 NHS Survey of Inpatients obtained information from 50% of all patients discharged between June and August. The 2010 RN4CAST England Nurse Survey gathered information from inpatient medical and surgical nurses.

**Main outcome measures** Patient ratings of their hospital care, their confidence in nurses and doctors and other indicators of their satisfaction. Missed nursing care was treated as both an outcome measure and explanatory factor.

**Results** Patients' perceptions of care are significantly eroded by lack of confidence in either nurses or doctors, and by increases in missed nursing care. The average number of types of missed care was negatively related to six of the eight outcomes—ORs ranged from 0.78 (95% CI 0.68 to 0.90) for excellent care ratings to 0.86 (95% CI 0.77 to 0.95) for medications completely explained—positively associated with higher patient-to-nurse ratios (b=0.15, 95% CI 0.10 to 0.19), and negatively associated with better work environments (b=−0.26, 95% CI −0.48 to −0.04).

**Conclusions** Patients' perceptions of hospital care are strongly associated with missed nursing care, which in turn is related to poor professional nurse (RN) staffing and poor hospital work environments. Improving RN staffing in NHS hospitals holds promise for enhancing patient satisfaction.

## INTRODUCTION

Highly publicised reports citing preventable deaths and deficiencies in hospital care in England have uniformly concluded that inadequate hospital professional nurse (RN) staffing is a contributing factor.[1–3] Studies confirm large variation in patient to RN ratios across National Health Service

### Strengths and limitations of this study

► This is the first quantitative study to determine the association between patients' confidence in nurses and doctors, RN staffing, and patient experiences with hospital care in National Health Service (NHS) hospitals in England using the national NHS Adult Inpatient Survey.
► Unique data previously unavailable enable a rigorous analysis of patient to RN staffing ratios, missed nursing care and patient satisfaction with hospital care.
► The study uses cross-sectional data, and while a number of alternative explanations are considered in our models, we cannot rule out the possibility that omitted variables contribute to associations found.

(NHS) hospitals, and this variation is associated with higher mortality in hospitals where RNs care for more patients each.[4–6] However, despite national guidance on safe nurse staffing,[7] substantial variation still exists and the value of higher RN staffing levels is still questioned at the policy level.[8] Recently introduced NHS workforce initiatives have been framed in the unsubstantiated narrative that quality deficiencies in hospitals are due to 'uncaring' nurses.[9 10] The National Advisory Group on the Safety of Patients in England specifically advised that nurses and other NHS staff not be blamed for quality deficits, pointing instead to the need to address insufficient RN staffing.[3] Nevertheless, new workforce initiatives have been introduced by the NHS purportedly to produce more caring nurses. One such initiative creates a new provider category, the nursing associate, with substantially lower qualifications than RNs.[11] Adding lesser trained providers to the hospital workforce without adding more RNs results in eroding the nursing skill mix that evidence suggests is associated with higher mortality and lower patient satisfaction.[12] Also, the NHS is reinstating apprentice training for

BMJ

RNs,[13] in direct opposition to a major recommendation of the 2010 Prime Minister's Commission on the Future of Nursing and Midwifery in England[14] that all nursing education should take place in universities because evidence shows that hospitals with a higher proportion of bachelors-prepared nurses have significantly better patient outcomes.[6 15 16]

The concern about nurses being uncaring or lacking in compassion, and subsequent NHS nursing initiatives, have come about largely in response to case studies of poor care in a relatively small number of NHS trusts and anecdotal reports of patient dissatisfaction. Surprisingly, little use has been made of the NHS National Inpatient Survey of patients to inform strategies to improve care.[17] When initiated in 2001, England's annual national survey of patients following a hospital inpatient stay was the first in the world; it aimed to make the NHS more patient-centred and more responsive to patient feedback.[18] A report published in 2007 using the NHS Inpatient Survey found evidence that the experiences of staff working in the NHS mirrored the experiences of patients receiving care.[19] This is a worrisome finding given the evidence showing high nurse burnout and job dissatisfaction is common in NHS hospitals,[10 20] and that 85% of RNs in NHS hospitals report not being able to complete needed nursing care due to lack of time associated with high patient-to-nurse workloads.[21] Furthermore, missed nursing care associated with high patient-to-nurse workloads is associated with an increased risk of patient mortality following common surgical procedures in nine European countries including England.[22]

Studies of patients' experiences with inpatient care in the USA, another country with mandated hospital patient satisfaction surveys, reveal that better RN staffing is associated with higher overall patient ratings of their hospitals.[23 24] Missed nursing care is associated with less favourable patient satisfaction in the US hospitals[25] and in some European hospitals (not including England).[26] There are not comparable studies in England using the NHS National Inpatient Survey that could help determine whether better RN staffing and better clinical hospital work environments are associated with more favourable patient experience with hospital care.

This paper seeks to identify an action agenda that may hold promise for improving patients' experiences with hospital care in England. Specifically, we first provide evidence of the importance of RNs to patients using data from a large sample of patients in NHS hospitals in England to show how patients' experience with care is strongly related to their confidence in nurses as well as doctors, and their perceptions of whether there were enough nurses in their hospitals. We then use data from patients and nurses in a subset of these hospitals to show how lower nurse workloads and better nurse work environments are related to less missed nursing care and how, in turn, less missed nursing care is related to better patients' experience with their care.

## METHODS
### Data sources and samples
Patient survey data are from the 2010 NHS Survey of Inpatients, which gathered information from over 66 000 patients who were discharged from 161 acute and specialist NHS trusts in England.[27] Nurse survey data are from the 2010 RN4CAST-England study,[20 28] which gathered information from 2963 inpatient medical and surgical direct care RNs in a representative sample of 31 of the same 161 NHS trusts. These 31 trusts comprise 46 different hospitals from which 12 581 of the 66 348 patients surveyed were discharged. Of these 12 851 patients, 5311 were in general medicine or general surgery wards. The sample of hospitals in which nurses were surveyed, described elsewhere in detail,[21 28] was a stratified random sample selected to include teaching and non-teaching hospitals of different sizes in every geographic region of England. There are no remarkable differences between the sample of hospitals in which nurses were surveyed and the other hospitals participating in the NHS Survey of Inpatients, nor were there any differences in patient characteristics or responses between the full NHS survey and the 31 trusts studied, as noted in the (online supplementary appendix) . The response rate for the NHS patient survey was 50%. The response rate for the nurse survey was 37%. The nurse survey has good established predictive validity in previous research,[6 29] showing, for example, that nurses' reports of quality of care are closely associated with patient mortality derived from independent data sources.[30]

Patients were not participants in the initial design of the overall study, but were actively engaged in the development of measures of patients' experiences with care used in the study. The Picker Institute, developers of the NHS Adult Inpatient Survey, employed patient focus groups and cognitive interviews with patients during pilot testing. Patients were offered one page to describe what they thought of the inpatient questionnaire and which aspects of patient care were most important to them. The qualitative research did not identify major questions missing from the survey but it did lead to minor modifications that were incorporated.[31] Patients in our study are anonymous. We have a detailed plan to disseminate the study results through print, broadcast and social media in every participating country. The authors gratefully acknowledge the contributions of participating patients in the 'Acknowledgements' section.

### Analysis strategy
These data were used to undertake three distinct but related analyses. First, we use patient data from all 161 trusts to describe how patients rated their care, how their ratings varied depending on their perceptions of whether there were enough nurses on duty to provide needed care and how they were as much a function of their confidence in nurses as their confidence in doctors. We then used the nurse data from the 46 hospitals in the 31 trusts to describe the variation in RN staffing and hospital work

**Table 1** Patient reports about nurses and doctors, and the per cent indicating their care was 'excellent', based on their reports about doctors and nurses

| Patient survey question | | Per cent of patients in each response category | Per cent of patients in each response category indicating that their care was 'excellent' |
|---|---|---|---|
| Did you have confidence and trust in the doctors treating you? | Yes, always | 80.4 | 52.6 |
| | Yes, sometimes | 16.4 | 9.4 |
| | No | 3.1 | 3.4 |
| | Total | 100.0 | |
| Did you have confidence and trust in the nurses treating you? | Yes, always | 75.1 | 55.3 |
| | Yes, sometimes | 21.7 | 10.5 |
| | No | 3.2 | 2.8 |
| | Total | 100.0 | |
| Were there enough nurses on duty to care for you in the hospital? | Always or nearly always | 60.4 | 57.3 |
| | Sometimes | 29.5 | 26.7 |
| | Never or rarely | 10.1 | 14.1 |
| | Total | 100.0 | |

The numbers reported exclude a small number (<2%) of missing responses.
Source: Data are from the 2010 National Health Service Survey of Inpatients, which involved 66 348 patients discharged from 161 trusts in England.

environments, and then used least-squares regression models with and without control variables to show how lower RN staffing levels and poorer work environments are related to needed but missed nursing care. Finally, since patient survey data were only available at the trust level, we merged the nurse data from the 31 trusts with patient data from those same trusts and used logistic regression models to estimate whether and to what extent the overall level of missed nursing care in the different trusts affect patients' ratings of their care and their confidence in nurses, before and after controlling for potential confounds. Because the nurse survey was restricted to nurses on medical and surgical units, this final step of the analysis was restricted to patients in general surgical and medical wards (5311 out of 12 851 patients in the study trusts).

## RESULTS

### Nurses, doctors and patient ratings of care

Table 1 and figure 1 use data from 66 348 patients in 161 trusts collected in the 2010 NHS Survey of Inpatients to show how patients' ratings of their care are highly associated with their confidence in nurses and in doctors, and with their perceptions of whether there were enough nurses to provide needed care. The first column of table 1 shows that more than three-fourths of patients responding to the NHS survey reported having confidence and trust in the doctors and nurses treating them, while only 60% reported that there were always or nearly always enough nurses to care for them. The second column of table 1 shows the percentages of patients that rated their care as excellent, based on how much confidence and trust

they had in their nurses and doctors, and their perceptions of the adequacy of the number of nurses caring for them. Hospital care was rated excellent by over half of the patients who indicated that they always had confidence and trust in their doctors or confidence and trust in their nurses, but by only 3% of the patients who never had confidence and trust in their doctors or in their nurses. Similarly, hospital care was rated as excellent by over half of the patients who indicated that there were always enough nurses to care for them, but by far lower

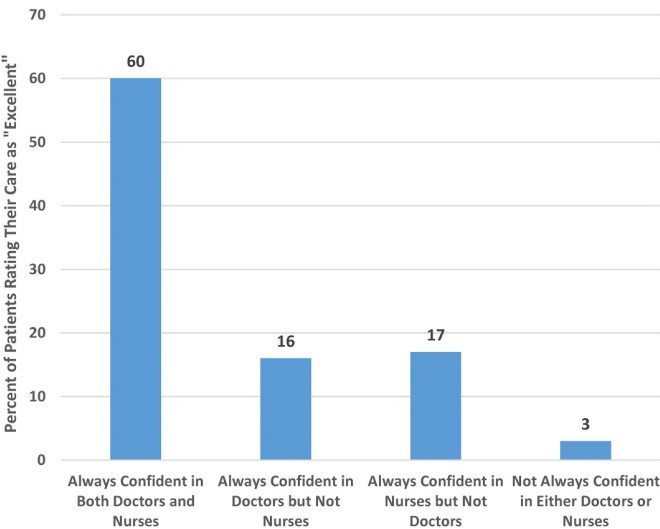

**Figure 1** Per cent of patients rating their care 'excellent', by confidence and trust in nurses and doctors. Source: Data are from the 2010 National Health Service Survey of Inpatients, which involved 66 348 patients discharged from 161 trusts in England.

percentages of patients who said there were only some-times enough, or rarely or never enough, nurses.

While table 1 makes it clear that nurses, like doctors, are importantly related to patients' perceptions of the quality of their care, figure 1 shows more directly that confidence and trust in nurses is of similar importance to confidence and trust in doctors. In figure 1, we show the per cent of patients that rated their care 'excellent' after grouping the patients into four categories, to distinguish patients who always have confidence and trust in both doctors and nurses, in doctors but not nurses, nurses but not doctors and neither doctors not nurses. Sixty per cent of patients who have confidence and trust in both doctors and nurses rate their care as excellent, while only 3% of patients who have confidence and trust in neither rate their care as excellent. When confidence and trust in either group erodes, the result is virtually identical. Only 16% of the patients who have confidence and trust in their doctors but not nurses rate their care as excellent, and only 17% of the patients who have confidence and trust in their nurses but not doctors rate their care as excellent.

### Nurse staffing, work environments and missed nursing care

Nurse (RN) staffing was estimated for the 46 hospitals included in RN4CAST-England by average nurse workloads in each hospital on the day shift. Nurses reported how many patients they cared for on their last shift, and then responses are averaged across all nurses in each hospital working the day shift. Nurse workloads averaged 8.6 patients per RN during the day, and ranged from 5.6 patients per RN to 11.5 patients per RN across the 46 hospitals. Patient-to-RN ratios were much higher and more variable at night so we elected to use day shift staffing only in our analyses.

Hospital work environment was measured by the Practice Environment Scale of the Nursing Work Index, an extensively used survey-based measure with established reliability and validity[32–35] leading to its adoption by the National Quality Forum as a nurse sensitive quality of care indicator.[36] The measure of work environment used is a composite measure formed from five subscales (comprising 28 nurse survey items) measuring resource adequacy (four items), nurse participation in hospital affairs (eight items), nursing foundations for quality care (nine items), nurse manager ability, leadership and support of nurses (four items) and nurse-physician relations (three items). The staffing and resource adequacy subscale was dropped from the global measure used in the analysis because of its high correlation with the direct measure of RN staffing in the model, as in previous publications.[12 37]

What makes the variability in staffing and work environments across hospitals of considerable importance is that when RNs have high patient loads, and when RNs practice in poor work environments, necessary nursing care can be missed because of lack of time.[21] Nurses in this study were asked whether any of 13 important types of nursing care were needed but missed because of lack

of time. Figure 2 shows that while 7% of nurses reported that they lacked time to complete necessary pain management, and 11% missed treatments and procedures, much greater percentages reported lacking the time to educate patients and their families (52%) and comfort or talk with their patients (65%). More than a quarter of the nurses (27%) lacked the time to complete three or four of the types of care listed, just under one in five (19%) lacked the time to complete five or six of them, and another 19% lacked the time to complete seven or more of the 13 types of care listed.

Table 2 provides regression coefficients that indicate the effects of RN staffing and the hospital work environment on the average number of types of missed care, before and after controlling for various hospital characteristics (including size, technology and location), and characteristics of nurses that may have affected their reports of missed care, including their role (primary nurse or shared responsibility for group of patients with other nurses), full-time status, years of experience and unit type (medical, surgical or combined). Higher nurse workloads (higher patient-to-RN ratios) are significantly related to higher numbers of types of missed care, while better work environments are significantly related to fewer types of missed care, both before and after adjustment.

Figures 3 and 4 show how much the number of tasks left undone varies as a function of RN staffing and hospital work environments, as estimated from the adjusted models. As the number of patients per RN goes down, from 12 patients to 8 patients to 4 patients, the average number of types of missed care goes down, from 4.4 (out of 13) to 3.8 to 3.2. And, as hospital work environments improve, from relatively poor (lowest tertile) to average (middle tertile) to relatively good (highest tertile), the average number of types of missed care also goes down, from 4.2 to 4.0 to 3.7.

### Missed nursing care and patient outcomes

The association of the number of types of missed care with patient outcomes is shown in table 3. The coefficients in the table are ORs which indicate how much the odds on providing a positive response to the nine different dimensions of patient satisfaction go down as the average number of types of missed care goes up, both before (unadjusted) and after (adjusted) taking account patient characteristics that might affect their responses, including gender, age, length of stay, ward, number of long-standing conditions and type of admission (emergent/urgent or planned). In all cases, the ORs are <1, indicating that positive patient appraisals of care decrease as the number of types of missed care increases; in six of the eight aspects of patient care rated the ORs are significant, and range from 0.78 to 0.86. These values indicate, for example, that in hospitals in which the number of types of missed care averaged 4.5 per nurse per shift, the odds on patients rating care as excellent and responding that the purpose of medicines were completely explained were 22% lower and 14% lower, respectively, than in

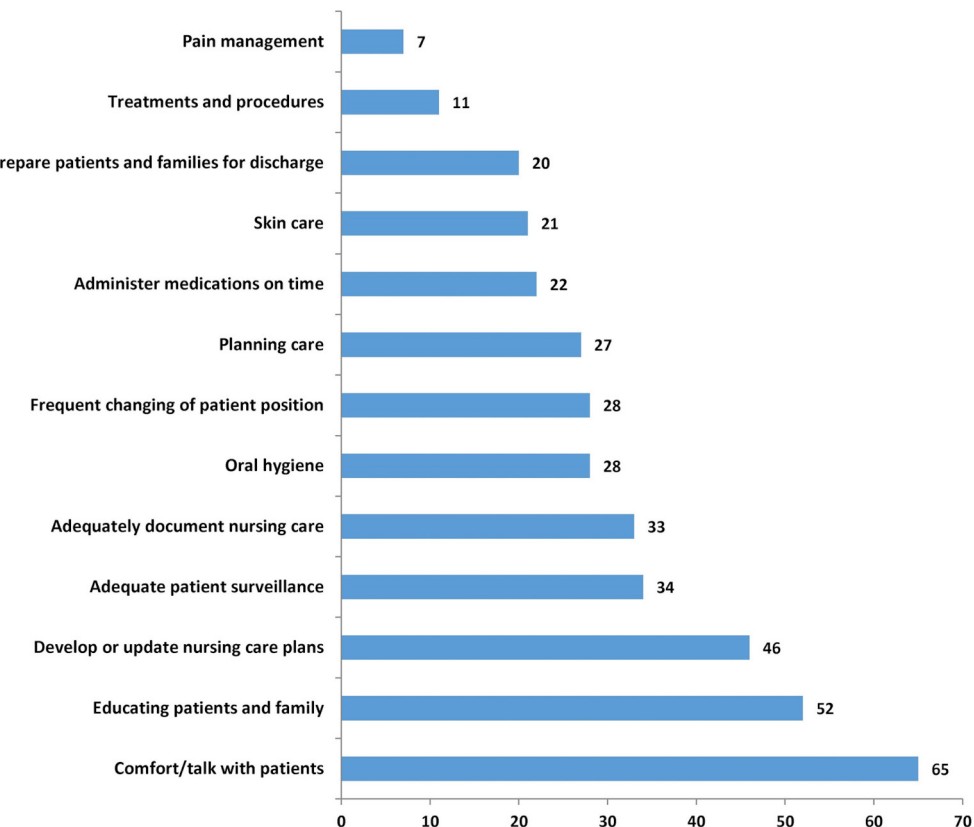

**Figure 2** Per cent of nurses reporting that different types of care were missed on their last shift. Source: Data are from the 2010 RN4CAST-England study, which surveyed 2963 inpatient medical and surgical direct care professional nurses (RNs) in a representative sample of 31 National Health Service trusts comprising 46 different hospitals.

hospitals in which the number of types of missed care averaged 3.5 per nurse per shift.

## DISCUSSION

National survey data from patients experiencing a hospitalisation in an NHS hospital in England confirm that

**Table 2** Regression coefficients indicating the effects of staffing and practice environment on average number of types of care missed

| Effect on missed care of | Regression coefficients (95% CI) | |
|---|---|---|
| | **Unadjusted** | **Adjusted** |
| Patient-to-nurse ratio | 0.11*** (0.06 to 0.16) | 0.15*** (0.10 to 0.19) |
| Practice environment | −0.30* (−0.55 to 0.05) | −0.26* (−0.48 to −0.04) |

Adjusted coefficients and CIs are from regression models which control for hospital characteristics (beds >750, high technology and location) and nurse characteristics (nurse role, full-time status, years of experience and unit type). Practice environment is measured by the Practice Environment Scale of the Nursing Work Index tertile.
Source: Data are from the 2010 RN4CAST-England study, which surveyed 2963 inpatient medical and surgical direct care professional nurses (RNs) in a representative sample of 31 National Health Service trusts comprising 46 different hospitals.
*P<0.05, **P<0.01, ***P<0.001.

patients have a high level of trust and confidence in RNs, evidence that refutes the narrative blaming quality of care deficits in NHS hospitals on uncaring nurses. However, only 60% of patients indicated that there were always enough RNs to care for them, and 1 in 10 patients indicated that there were never or rarely enough RNs. The importance to patients of adequate RN staffing is evident in their responses; 57% of patients who indicated that there were always or nearly always enough RNs to care for them rated care as excellent, compared with only 14% of the patients who said there were rarely or never enough. Additional analyses undertaken (not shown) indicate that

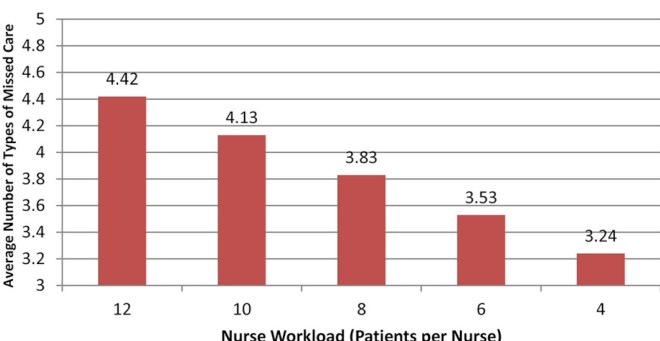

**Figure 3** Average number of types of missed care, by nurse workload. Source: Data are from the 2010 RN4CAST-England study, which surveyed 2963 inpatient medical and surgical direct care professional nurses (RNs) in a representative sample of 31 National Health Service trusts comprising 46 different hospitals.

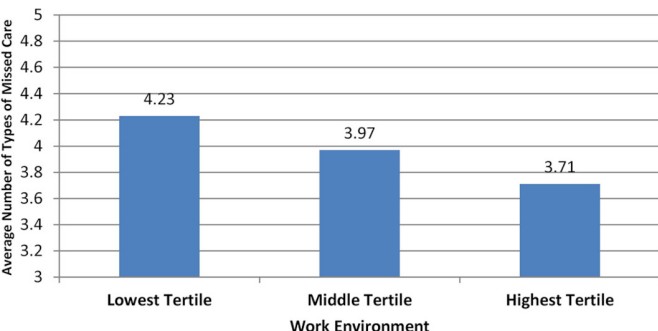

**Figure 4** Average number of types of missed care, by work environment. Source: Data are from the 2010 RN4CAST-England study, which surveyed 2963 inpatient medical and surgical direct care professional nurses (RNs) in a representative sample of 31 National Health Service trusts comprising 46 different hospitals.

**Table 3** ORs indicating the effect of the trust median number of types of care missed on various patient outcomes

| Effect of the median number of types of care missed on patient outcomes | ORs (95% CIs) | |
|---|---|---|
| | **Unadjusted** | **Adjusted** |
| Rate care excellent | 0.79*** | 0.78*** |
| | (0.69 to 0.90) | (0.68 to 0.90) |
| Did not want to complain about care | 0.92 | 0.92 |
| | (0.76 to 1.12) | (0.77 to 1.11) |
| Always felt treated with respect and dignity | 0.89 | 0.92 |
| | (0.78 to 1.02) | (0.81 to 1.06) |
| Completely explained purpose of medicines | 0.87* | 0.86** |
| | (0.78 to 0.98) | (0.77 to 0.95) |
| Doctors and nurses work together excellent | 0.84** | 0.82** |
| | (0.74 to 0.94) | (0.72 to 0.93) |
| Always got answers I could understand | 0.84** | 0.83*** |
| | (0.75 to 0.95) | (0.76 to 0.91) |
| Always have confidence and trust in nurses | 0.86* | 0.85* |
| | (0.74 to 0.99) | (0.73 to 0.99) |
| Always or nearly always enough nurses | 0.87* | 0.85** |
| | (0.76 to 0.99) | (0.75 to 0.96) |

Adjusted models control for hospital characteristics (beds >750, high technology and location) and patient characteristics that might affect responses, including gender, age, length of stay, ward, number of long-standing conditions and type of admission (emergent/urgent or planned).
Source: Data are from a merged file that included information from 31 NHS trusts for which both patient information (from 5311 general medical and surgical patients included in the 2010 NHS Survey of Inpatients) and nurse information (from 2963 medical and surgical nurses surveyed in the 2010 RN4CAST-England study) were available.
NHS, National Health Service.
*P<0.05, **P<0.01, ***P<0.001.

patients in hospitals with poorer RN staffing are much less likely to say there were always enough nurses to care for them. We estimate, from models that took account of numerous confounds, that the likelihood of patients saying there were always enough nurses to take care of them were about 40% lower in hospitals in which the average nurse took care of 10 patients than in hospitals in which the average nurse took care of 6 patients. These findings reinforce from patients' perspectives the importance of adequate hospital RN staffing.

Further insights into how quality of care might be improved in NHS hospitals is revealed when data from the NHS Inpatient Survey is linked with information on actual hospital RN staffing and nurses' assessments of the quality of their work environments. We found substantial variation across NHS general acute hospitals in patient-to-nurse workloads. Nurses in some NHS hospitals are caring for twice as many patients at a time as nurses in other hospitals. Current NHS policies devolving greater autonomy to hospital management to make decisions about RN staffing may be contributing to the substantial observed variation in staffing, and have led experts to point to the need for checks and balances to minimise the risk of more quality failures linked to inadequate RN staffing.[38] Our findings show that the substantial differences in RN staffing across NHS hospitals are associated with the extent to which needed nursing care is missed. The most frequently missed types of care include those that patients may readily recognise are missing–comforting and talking with patients, and teaching patients and family members how to manage care following discharge. Our results are consistent with other research showing that higher patient workloads for RNs in NHS hospitals are associated with adverse patient outcomes including higher hospital morality.[4 5 22] Initiatives such as those recently adopted in Wales[39] establishing an upper limit to how many patients nurses can safely and effectively care for holds promise for further improvements in patients' satisfaction with hospital care, and may save lives as well.

Another modifiable feature of hospital care found to be relevant to patients' perceptions of their care is the quality of the hospital work environment. In hospitals rated by nurses to have less favourable clinical work environments, needed but missed nursing care is more extensive. Patients' perceptions of care are less favourable when missed care is more extensive. Research suggests that hospital work environments that support RNs to provide care efficiently and effectively, and without constant interruptions because of operational failures such as missing medications and equipment,[40] are reasonably low cost interventions and return good value in terms of better patient outcomes at the same or lower costs.[41 42] Magnet hospitals formally recognised for their good hospital work environments have significantly higher patient satisfaction than matched non-Magnet hospitals.[43] One of the first Magnet hospitals accredited outside the USA was an NHS trust in England, which research showed

significantly improved its work environment and care quality during the process of achieving Magnet accreditation.[44] Unfortunately, the NHS merged the Magnet facility out of existence after a year, and there has not been a Magnet-accredited hospital in England in over 15 years.

Patients' confidence in both doctors and nurses is equally important in how patients rate their hospitals; few patients who have high confidence in their doctors but little confidence in their nurses rate their hospitals highly. This finding is relevant to policy decisions governing the composition of the NHS workforce in England. Between 2010 and 2015, the number of physician consultants (mostly inpatient physicians) increased by more than one-fifth while the number of RNs increased by only 1%.[45]

Our study has many strengths including use of validated measures of patient satisfaction, nurse staffing, hospital work environment and missed nursing care across large numbers of NHS hospitals. The study has limitations as well. Data from both patients and nurses are cross-sectional, thus limiting causal inferences about the associations found. We take into account, to the extent possible, alternative explanations about factors that could be associated with our findings including characteristics of hospitals such as teaching status, and characteristics of the patients responding to the national survey, including their health status since self-reported limiting long-term conditions have been found to be associated with less favourable perceptions of care.[46] Our data are from 2010 but remain the only comprehensive data on hospital nurse workforce and patient satisfaction across large numbers of NHS hospitals in England. Moreover, our interest is in the relationship between patient satisfaction and nurse resources, and there is no reason to expect the relationship to have changed since 2010. Indeed, Sir Robert Francis, author of the public inquiry into quality of care deficiencies at the Mid Staffordshire NHS Trust,[1] commented as recently as July 2017 that safe nurse staffing in England still lacks a standardised approach and substantial variation across hospitals in nurse staffing remains.[47]

## CONCLUSIONS

Patients express a high level of confidence and trust in nurses, and their satisfaction with hospital care is less favourable when they perceive there are not enough nurses available. The narrative that quality deficits in hospitals in England are due to 'uncaring' nurses is not supported by the evidence. On the contrary, our findings suggest that reducing missed nursing care by ensuring adequate numbers of RNs at the hospital bedside and improved hospital clinical care environments are promising strategies for enhancing patient satisfaction with care.

**Acknowledgements** The authors thank Tim Cheney for analytic assistance, the patients, nurses and hospitals that participated in the study and the RN4CAST Consortium.

**Contributors** LHA raised the funding, developed the idea for the study, interpreted results, drafted and revised manuscript; DMS conducted the analysis, drafted the results, contributed to manuscript revisions; JB collected survey data, contributed to manuscript development; LB contributed to results interpretation and manuscript drafting and revisions; AMR contributed to data collection, interpretation of results, manuscript revisions; PG contributed to obtaining data, results interpretation, manuscript development and revisions.

**Funding** European Union's Seventh Framework Program (223468) and the National Institute of Nursing Research, National Institutes of Health (R01 NR014855).

**Competing interests** None declared.

**Ethics approval** University of Pennsylvania Institutional Review Committee (IRB).

**Provenance and peer review** Not commissioned; externally peer reviewed.

**Data sharing statement** No additional data are available.

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
