## [Reviewer comments · BMJ Open]

ARTICLE DETAILS

TITLE (PROVISIONAL)	Patient Satisfaction with Hospital Care and Nurses in England: An Observational Study
AUTHORS	Aiken, Linda; Sloane, Douglas; Ball, Jane; Bruyneel, Luk; Rafferty, Anne Marie; Griffiths, Peter

VERSION 1 – REVIEW

REVIEWER	JÖRG KLEWER Faculty of Public Health and Nursing Science University of Applied Sciences Zwickau Germany
REVIEW RETURNED	22-Aug-2017

GENERAL COMMENTS	It is an interesting paper, which should be accepted for publication. Before publication, some minor revisions are recommended: The last paragraph of the introduction presents methodological information, that is why these information should be integrated in the following chapter (methods). On the other hand, a detailed description of the objectives should be inserted at the end of the introduction. At time, the reader hat to read a little bit between the lines to reveal the objectives. Results: Especially at the beginning table 1 is described very thoroughly. Normally the chapter results should present the most important results, merely referring to the tables/ figures (e.g. (table 1)). Tables should be created in a way that enables to understand them without long explanations written in the text. Therefore some of the paragraphs should be rewritten, so that the reader is able to realize the main findings without looking at the tables. Figure 3 & 4: the labels of the Y-axis are missing.
---

REVIEWER	Damien Contandriopoulos Université de Montréal (Canada)
REVIEW RETURNED	28-Aug-2017

GENERAL COMMENTS	This paper links data from two sources to study the interrelations between 1) patients' satisfaction / perceived quality 2) patients' confidence in nurses and doctors 3) nurse to patient ratio and 4) nurses' perceived quality of work environment.
--

The analyses conducted lead to three main results

A) Patient satisfaction is highly associated with the trust patients have in their nurses and doctors and in their perception that there were enough nurses to provide needed care.

Regarding this result, although the data used to show this is new, to me it isn't new or surprising that patients have a high level of trust and confidence in RNs. As I suggest below, for me the paper's core contribution is the point C and maybe it would make sense to focus more on it and less on point A.

B) According to the paper, the trust in nurses is as important as the trust in doctors to explain patient satisfaction (p. 6 lines 24 and down). However, I'm not completely convinced that Figure 1 really shows "that confidence and trust in nurses is of equal importance to confidence and trust in doctors". To me what the figure shows is that most patients had trust in both doctors and nurses. When this was not the case, usually patients still trusted one of the two professional groups. Rarely did patients trust no-one (luckily). As for the relative importance of trust in doctors or nurses, the figure says nothing. If there is something that I missed here, maybe the argument could be made clearer.

C) As the average number of (nurses reported) missed care increases different dimensions of the (patients reported) quality of care decreases. The association between nurses-reported quality of work environment and patient-reported quality of care is, in my view, the most important aspect of this paper. However, it is somewhat buried in the arguments and discussions of other interrelations between variables. In my view, a clearer presentation of this association and why it matters could strengthen the article usefulness for practice.

Regarding this link, the correlation between actual nurse to patient ratio and % of patients indicating that there were always enough RNs to care for them would be very interesting. I was left a bit puzzled by the fact it appears the authors have the two variables but do not provide an analysis of their association. If positive, such an analysis would greatly strengthen the argument that low ratios -> quality of care -> patients' perceptions -> satisfaction.

Odd details:

p. 2 line 40: The sentence hints at a causal link (Better RN staffing causes higher ratings) although the causal relation behind the correlation is unclear

p. 2 line 49: Last sentence on the page is unclear. Nursing being either the problem or solution ?

p. 3 line 3: First sentence of the page. The formulation stating that "patient satisfaction is a function of" is weird.

VERSION 1 – AUTHOR RESPONSE

Reviewer: 1

Reviewer Name: JÖRG KLEWER

Institution and Country: Faculty of Public Health and Nursing Science, University of Applied Sciences, Zwickau, Germany

Please state any competing interests: None declared

It is an interesting paper, which should be accepted for publication.

Before publication, some minor revisions are recommended:

Comment: The last paragraph of the introduction presents methodological information, that is why these information should be integrated in the following chapter (methods). On the other hand, a detailed description of the objectives should be inserted at the end of the introduction. At time, the reader had to read a little bit between the lines to reveal the objectives.

Response: We deleted the methodological information from the last paragraph of the introduction. The same information (in greater detail) was already provided in the Methods section, and remains there. We also inserted a clearer statement of the objectives at the end of the introduction.**

Results

Comment: Especially at the beginning table 1 is described very thoroughly. Normally the chapter results should present the most important results, merely referring to the tables/ figures (e.g. (table 1)). Tables should be created in a way that enables to understand them without long explanations written in the text. Therefore some of the paragraphs should be rewritten, so that the reader is able to realize the main findings without looking at the tables.

Response: We edited the text to shorten our explanation of the results in Table 1 and to make those results understandable to the reader without reference to the table.**

Comment: Figure 3 & 4: the labels of the Y-axis are missing.

Response: We added labels to the vertical axes in both figures.**

Reviewer: 2

Reviewer Name: Damien Contandriopoulos

Institution and Country: Université de Montréal (Canada)

Please state any competing interests: None declared

Please leave your comments for the authors below

This paper links data from two sources to study the interrelations between 1) patients' satisfaction / perceived quality 2) patients' confidence in nurses and doctors 3) nurse to patient ratio and 4) nurses' perceived quality of work environment.

The analyses conducted lead to three main results

Comment A) Patient satisfaction is highly associated with the trust patients have in their nurses and doctors and in their perception that there were enough nurses to provide needed care.

Regarding this result, although the data used to show this is new, to me it isn't new or surprising that patients have a high level of trust and confidence in RNs. As I suggest below, for me the paper's core contribution is the point C and maybe it would make sense to focus more on it and less on point A.

Response: Regarding what the reviewer refers to as point A, a key intended audience for our paper is decision makers in the NHS who are citing and acting upon the unsubstantiated narrative that RNs in NHS hospitals are indifferent and uncaring toward patients. We offer in the paper several examples of recent workforce policies such as the introduction of nursing associates and a return to apprenticeship, rather than science-based, education for RNs that have been described by the NHS as strategies for recruiting "more caring" nurses to the bedside. While we appreciate that our finding that patients have a high level of trust and confidence in RNs will not surprise those in healthcare, we think that debunking that myth using data from the NHS National Inpatient Survey is relevant to informing evidence-based policymaking.

We agree with the reviewer's argument that our multivariate results which show that staffing and work environments are related to the average number of types of missed care, and that the average number of types of missed care is in turn related to patient reported quality of care (which the reviewer refers to as point C), is the core contribution here. And, we agree that it is deserving of greater focus than the simpler and largely bivariate findings that patient satisfaction is related to the trust patients have in their nurses and doctors and in their perception whether there were enough nurses to provide needed care (which the reviewer refers to as point A). The only thing that might provoke the sense that we focus more on point A than point C is that we introduce Point A first. But we cover point A using one table, one figure, 379 words of text in the Results, and 215 words of text in the Discussion section. We cover point C using two tables, three figures, 837 words of text in the Results, and 445 words of text in the Discussion section. Our primary reason for focusing on point A prior to Point C is because we wanted to establish that trust and confidence in nurses and patient perceptions of the adequacy of the supply of nurses are importantly connected to patients' experiences (or "satisfaction" as the reviewer describes it) before describing how fewer patients per nurse and better work environments diminish missed care and in so doing improve patients' experience. Moreover, as we describe clearly in the Data Sources and Samples section and in the Analysis Strategy section of the paper, we were able to bring data from over 66,000 patients from 161 NHS trusts to bear on point A, but data from only a subsample of 12,581 patients in 31 of the same trusts to bear on the final analyses related to point C. We understand the reviewer's point, but believe that the order in which the different analyses are shown is appropriate, and our "focus" is appropriately placed.**

Comment B) According to the paper, the trust in nurses is as important as the trust in doctors to explain patient satisfaction (p. 6 lines 24 and down). However, I'm not completely convinced that Figure 1 really shows "that confidence and trust in nurses is of equal importance to confidence and trust in doctors". To me what the figure shows is that most patients had trust in both doctors and nurses. When this was not the case, usually patients still trusted one of the two professional groups. Rarely did patients trust no-one (luckily). As for the relative importance of trust in doctors or nurses, the figure says nothing. If there is something that I missed here, maybe the argument could be made clearer.

Response: While the title to Figure 1 did make clear that what we were showing was the percent of patients that rated their care "excellent" after patients were grouped by their confidence in doctors and nurses, the fact that the vertical axis label read only "Percent" could have been confusing. We have titled the axis more appropriately to read "Percent of Patients Rating Their Care as Excellent", and edited the text to clarify this. Thus, the figure does say something about the relative importance of trust in doctors or nurses. As we appropriately conclude "When confidence and trust in either group erodes, the result is virtually identical.

Only 16% of the patients who have confidence and trust in their doctors but not nurses rate their care as excellent, and only 17% of the patients who have confidence and trust in their nurses but not doctors rate their care as excellent.” As noted in the discussion, this finding is relevant to informing policies to improve patient satisfaction in view of documented increases in investments in hospital physician staffing but almost no increase in RN staffing.**

Comment C) As the average number of (nurses reported) missed care increases different dimensions of the (patients reported) quality of care decreases. The association between nurses-reported quality of work environment and patient-reported quality of care is, in my view, the most important aspect of this paper. However, it is somewhat buried in the arguments and discussions of other interrelations between variables. In my view, a clearer presentation of this association and why it matters could strengthen the article usefulness for practice.

Response: We have edited the last sentence of the Introduction to clarify our objectives, which is to show “how lower nurse workloads and better nurse work environments are related to lower volumes of missed nursing care and how, in turn, less missed nursing care is related to better patients’ experience with their care.” We modeled these associations in two steps. We did this for several reasons. The NHS Inpatient Sample reports patient outcomes by trusts rather than hospitals. Our results showed substantial variation in our major variables of interest—nurse staffing and work environment—across hospitals within trusts so we were able to obtain better estimates of those factors on missed care than we could have on patients’ experience. Also we were underpowered to estimate the direct effects of nurse staffing and nurse work environments on patients’ experience, while simultaneously estimating their indirect effects through missed care with only 31 trusts (rather than for 46 hospitals) which all the requisite data were available.**

Comment: Regarding this link, the correlation between actual nurse to patient ratio and % of patients indicating that there were always enough RNs to care for them would be very interesting. I was left a bit puzzled by the fact it appears the authors have the two variables but do not provide an analysis of their association. If positive, such an analysis would greatly strengthen the argument that low ratios -> quality of care -> patients' perceptions -> satisfaction.

Response: For reasons noted immediately above, we did not attempt to estimate the effects of RN staffing (or the work environment) on the indicators of patient experience, of which “perceptions that there were always enough RNs to care for them” was one. In response to the reviewer’s query, the discussion was extended as follows:

“We undertook additional analyses that indicate that patients in hospitals with poorer staffing are much less likely to report there were always enough RNs to care for them. We estimate, from models that took account of numerous confounds, that the likelihood of patients reporting there were always enough nurses to take care of them were 40% lower in hospitals in which the average nurse took care of 10 patients than in hospitals in which the average nurse took care of 6 patients.”**

Odd details:

Comment: p. 2 line 40: The sentence hints at a causal link (Better RN staffing causes higher ratings) although the causal relation behind the correlation is unclear

Response: We have edited the sentence to read “Studies of patients’ experiences with inpatient care in the U.S., another country with mandated hospital patient satisfaction surveys, reveal that better RN staffing is associated with higher overall patient ratings of their hospitals”***

Comment: p. 2 line 49: Last sentence on the page is unclear. Nursing being either the problem or solution ?

Response: We have edited the sentence to read "There are not comparable studies in England using the NHS National Inpatient Survey that could help determine whether better nurse staffing and better nurse work environments are related to missed nursing care and, ultimately, to more favorable patient satisfaction in hospitals."**

Comment: p. 3 line 3: First sentence of the page. The formulation stating that "patient satisfaction is a function of" is weird.

Response: We have edited the last paragraph of the introduction in response to the first concern expressed by reviewer 1, and in so doing deleted this sentence.**

VERSION 2 – REVIEW

REVIEWER	JÖRG KLEWER University of Applied Sciences Zwickau Germany
REVIEW RETURNED	10-Sep-2017

GENERAL COMMENTS	No revisions required.
------------------------

REVIEWER	Damien Contandriopoulos Université de Montréal (Canada)
REVIEW RETURNED	12-Sep-2017

GENERAL COMMENTS	The revisions and answers provided are satisfactory
---